# VARIATIONAL INFERENCE FOR DIFFUSION MODULATED COX PROCESSES

## ABSTRACT

This paper proposes a stochastic variational inference (SVI) method for computing an approximate posterior path measure of a Cox process. These processes are widely used in natural and physical sciences, engineering and operations research, and represent a non-trivial model of a wide array of phenomena. In our work, we model the stochastic intensity as the solution of a diffusion stochastic differential equation (SDE), and our objective is to infer the posterior, or smoothing, measure over the paths given Poisson process realizations. We first derive a system of stochastic partial differential equations (SPDE) for the pathwise smoothing posterior density function, a non-trivial result, since the standard solution of SPDEs typically involves an Itô stochastic integral, which is not defined pathwise. Next, we propose an SVI approach to approximating the solution of the system. We parametrize the class of approximate smoothing posteriors using a neural network, derive a lower bound on the evidence of the observed point process sample-path, and optimize the lower bound using stochastic gradient descent (SGD). We demonstrate the efficacy of our method on both synthetic and real-world problems, and demonstrate the advantage of the neural network solution over standard numerical solvers.

## 1 INTRODUCTION

*Cox processes* (Cox, 1955; Cox & Isham, 1980), also known as *doubly-stochastic Poisson processes*, are a class of stochastic point processes wherein the point intensity is itself stochastic and, conditional on a realization of the intensity process, the number of points in any subset of space is Poisson distributed. These processes are widely used in the natural and physical sciences, engineering and operations research, and form useful models of a wide array of phenomena.

We model the intensity by a diffusion process that is the solution of a stochastic differential equation (SDE). This is a standard assumption across a range of applications (Susemihl et al., 2011; Kutschireiter et al., 2020). The measure induced by the solution of the SDE serves as a prior measure over sample paths, and our objective is to infer a posterior measure over the paths of the underlying intensity process, given realizations of the Poisson point process observations over a fixed time horizon. This type of inference problem has been studied in the Bayesian filtering literature (Schuppen, 1977; Bain & Crisan, 2008; Särkkä, 2013), where it is of particular interest to infer the state of the intensity process at any past time given all count observations till the present time instant (the resulting posterior is called the smoothing posterior measure).

In a seminal paper, Snyder (1972) derived a stochastic partial differential equation (SPDE) describing the dynamics of the corresponding posterior density for Cox processes. The solution of this *smoothing SPDE* requires the computation of an Itô stochastic integral with respect to the counting process. It has long been recognized (Clark, 1978; Davis, 1981; 1982) that for stochastic smoothing (and filtering) theory to be useful in practice, it should be possible to compute smoothing posteriors conditioned on a single observed sample path. However, Itô integrals are not defined pathwise and deriving a *pathwise* smoothing density is remarkably hard. 30 years after Synder's original work Elliott & Malcolm (2005) derived a pathwise smoothing SPDE in the form of a coupled system of forward and backward pathwise SPDEs. Nonetheless, solving the system of pathwise SPDEs, or sampling from the corresponding SDE, is still challenging and intractable in general. It is well known, for example, that numerical techniques for solving these SPDEs, such as the finite element

method (FEM), suffers from the curse of dimensionality (Han et al., 2018). Therefore, it is of considerable interest to find more efficient methods to solve the smoothing SPDE.

We take a variational inference approach to computing an approximate smoothing posterior measure. Variational representations of Bayesian posteriors in stochastic filtering and smoothing theory have been developed in considerable generality; see (Mitter & Newton, 2003) for a rigorous treatment. There are a number of papers that consider the computation of an approximate posterior distribution over the paths of the underlying intensity process that is observed with additive Gaussian noise (Archambeau et al., 2007; 2008; Cseke et al., 2013; Susemihl et al., 2011; Sutter et al., 2016). Susemihl et al. (2011) studied Bayesian filtering of Gaussian processes by deriving a differential equation characterizing the evolution of the mean-square error (MSE) in estimating the underlying Gaussian process. On the other hand, Sutter et al. (2016) compute a variational approximation to the smoothing posterior density when the underlying diffusion intensity is observed with additive Brownian noise. They choose their variational family to be a class of SDEs with an analytically computable marginal density. This setting is considerably different from our setting, where the observed process is a point process. Nonetheless, Sutter et al. (2016) provides methodological motivation for our current study. In the context of the computation of approximate smoothing/filtering posteriors for point process observations, Harel et al. (2015) developed an analytically tractable approximation to the filtering posterior distribution of a diffusion modulated marked point processes under specific modeling assumptions suited for a neural encoding/decoding problem. In general, however, analytical tractability cannot be assured without restrictive assumptions.

We present a stochastic variational inference (SVI) (Hoffman et al., 2013) method for computing a variational approximation to the smoothing posterior density. Our approach fixes an approximating family of path measures to those induced by a class of parametrized SPDEs. In particular, we parametrize the drift function of the approximating SPDEs by a neural network with input and output variables matching the theoretical smoothing SPDE. Thereafter, using standard stochastic analysis tools we compute a tractable lower bound to the evidence of observing a sample path of count observations, the so-called *evidence lower bound* (ELBO). A sample average approximation (SAA) to the ELBO is further computed by simulating sample paths from the stochastic differential equation (SDE) corresponding to the approximating SPDE. Finally, by maximizing the ELBO, the neural network is trained using stochastic gradient descent (SGD) utilizing multiple batches of sample paths of count observations. Note that each sample path of the count observations entails the simulation of a separate SDE. We note that there are many problems in the natural and physical sciences, engineering and operations research where multiple paths of a point process (over a finite time horizon) may be obtained. For instance, we present an example in Section 5 modeling the demand for bikes rented during a 24 hour, one day time period in a bike-sharing platform, where the underlying driving intensity is subject to stochastic variations, and demand information is collected over multiple days.

In contrast to the variational algorithm developed in Sutter et al. (2016), where the variational lower bound must be re-optimized for new sample paths of the observation process, our variational method is more general and our approximation to the smoothing posterior can be used as a map for another (unobserved) sample path of count observations. Our computational approach can also be straightforwardly adapted to solve the problem of interest in Sutter et al. (2016).

In the subsequent sections, we describe our problem and method in detail and demonstrate the utility of our method with the help of numerical experiments. In particular, we show how the choice of approximating family enables us to use the trained neural network and in turn, the variational Bayesian smoothing posterior (VBSP), to compute smoothing SPDE in almost $(3/4)^{th}$ of the computational time required to compute the original smoothing SPDE using FEM. Moreover, we also efficiently generate Monte Carlo samples from the learned VBSP and use them for inference on the bike-sharing dataset, whereas FEM failed to compute either VBSP or the true smoothing density for the given time-space discretization.

## 2 PROBLEM DESCRIPTION

Let $\mathbf{N}_t$ be a Cox process with unknown stochastic intensity $\{z_t \in \mathbb{R}^+, t \in [0, T]\}$. We use $N_{t', t}$ to denote a sample path realization of $\mathbf{N}_t$ restricted to the interval $[t', t]$, and use $N_t$ to denote $N_t - N_0$; recall that $\mathbf{N}_0 = 0$ by definition. As noted before, a Cox process conditioned on the intensity is a

Poisson process. Therefore, given a realized sample path $\{z_t, t \in [0, T]\}$ of the intensity, and for any $0 \le t' < t \le T$, the marginal likelihood of observing $N_t - N_{t'} \in \mathbb{N}$ counts in $(t', t]$ is

$$\mathbf{N}_t - \mathbf{N}_{t'} \sim \mathcal{L}(\mathbf{N}_t - \mathbf{N}_{t'} = N_t - N_{t'} | \{\boldsymbol{z}_s\}_{t' < s \le t}) := \left( \int_{t'}^{t} \boldsymbol{z}_s ds \right)^{N_t - N_{t'}} \frac{e^{-\int_{t'}^{t} \boldsymbol{z}_s ds}}{(N_t - N_{t'})!}, \quad (1)$$

where $\mathcal{L}$ denotes the Poisson likelihood. Rather than directly modeling the intensity $\boldsymbol{z}$, we will bring a little more flexibility to our setting, and assume that $\boldsymbol{z}_t$ is a deterministic transformation of an another stochastic process $\boldsymbol{x}_t$ through a known mapping $h : \mathbb{R}^d \mapsto \mathbb{R}^+$: is $\boldsymbol{z}_t = h(\boldsymbol{x}_t)$. Note that the non-negative range of $h$ ensures that the Poisson intensity $\boldsymbol{z}_t = h(\boldsymbol{x}_t)$ is non-negative. Unless $\boldsymbol{x}_t \in \mathbb{R}^+$, the mapping $h$ cannot be an identity function. We use the term intensity process to refer to either $\boldsymbol{z}_t$ or $\boldsymbol{x}_t$.

We model the intensity process $\{\boldsymbol{x}_t \in \mathbb{R}^d, \forall t \in [0, T]\}$ with the following SDE,

$$d\boldsymbol{x}_t = b(\boldsymbol{x}_t)dt + \sigma(\boldsymbol{x}_t)d\mathbf{B}_t, \forall t \le T \text{ and } \boldsymbol{x}_0 = \mathbf{0}, \quad (2)$$

where $b : \mathbb{R}^d \mapsto \mathbb{R}^d$ is the drift function, $\sigma(\cdot) : \mathbb{R}^d \mapsto \mathbb{R}^{d \times d}$ is the diffusion coefficient, and $\mathbf{B}_t$ is the $d-$dimensional Brownian motion (or Wiener process). We assume that there exists a strong solution to the SDE above (Oksendal, 2013, Chapter 5). Moreover, we assume that $b(\cdot)$, $h(\cdot)$, and $\sigma(\cdot)$ are fixed by the modeler apriori, and we are interested in inferring the unknown intensity process with their fixed definitions. Incorporating them will obscure our main contribution, and we leave it for future work.

The model of the count observations above forms a *diffusion modulated Cox process*. Diffusion modulated Cox processes are widely used to model the arrival process in various service systems such as call centers, hospitals, airports etc. (Zhang et al., 2014; Wang et al., 2020). Zhang & Kou (2010) use a Gaussian process modulated Cox-process to infer proteins' conformation, in particular, they model the arrival rates of the photons collected from a laser excited protein molecule as a Gaussian process. Schnoerr et al. (2016) model spatio-temporal stochastic systems from systems biology and epidemiology using Cox process where intensity is modelled with diffusions.

As stated in the introduction, we seek to infer the smoothing posterior measure over the unknown intensity process $\{\boldsymbol{x}_t, t \in [0, T]\}$ using the count observations upto time $T$. Following terminology from the Bayesian filtering theory (Särkkä, 2013), we use *smoothing* to refer to inferring the unobserved intensity process at any past time given the observations upto the current time. Mathematically, the smoothing posterior is defined using the conditional expectation of the form $\mathbb{E}[f(\boldsymbol{x}_t) | \boldsymbol{s}(\mathbf{N}_u, u \in [0, T])]$, where $\boldsymbol{s}(\mathbf{N}_u, u \in [0, T])$ is the smallest sigma algebra (or filtration) generated by the Cox process $\{\mathbf{N}_t\}$ from time $0$ to $T$. For brevity we write $\mathbb{E}[f(\boldsymbol{x}_t) | \boldsymbol{s}(\mathbf{N}_u, u \in [0, T])]$ as $\mathbb{E}[f(\boldsymbol{x}_t) | N_{0,T}]$. Interested readers may refer to Kutschireiter et al. (2020) for more details on non-linear filtering theory.

We now provide a formal derivation of the smoothing posterior using Bayes' theorem (Bain & Crisan (2008); Elliott & Malcolm (2005)). Observe the conditional expectation satisfies

$$\mathbb{E}[f(\boldsymbol{x}_t) | N_{0,T}] = \frac{\mathbb{E}^{\dagger}[\Lambda_{0,T} f(\boldsymbol{x}_t) | N_{0,T}]}{\mathbb{E}^{\dagger}[\Lambda_{0,T} | N_{0,T}]} \quad (3)$$

for any measurable function $f(\cdot)$ and $\Lambda_{s,t} := \frac{\mathcal{L}(N_{s,t})}{\mathcal{L}^{\dagger}(N_{s,t})}$ for any $0 \le s < t \le T$, where $\mathcal{L}^{\dagger}$ is the unit intensity Poisson likelihood and $\mathbb{E}^{\dagger}[\cdot]$ denotes the expectation with respect to $\mathcal{L}^{\dagger}$. Note that $\mathcal{L}^{\dagger}$ does not depend on the stochastic intensity process $\boldsymbol{x}$ and forms a reference measure. The *marginal smoothing posterior density* is defined as

$$p_t(x | N_{0,T}) := \mathbb{P}(\mathbf{x}_t \in dx | N_{0,T}), \quad (4)$$

which can be formally obtained from equation 3 by setting $f(\boldsymbol{x}_t) = \mathbb{I}_{\{A\}}(\boldsymbol{x}_t)$ for any $A \in \mathbb{R}^d$, where $\mathbb{I}_{\{A\}}(y)$ is an indicator function that equals 1 when $y \in A$, otherwise 0. Now, define the unnormalized *filtering* density function $\bar{q}_t(x)$ as the function satisfying

$$\mathbb{P}(\boldsymbol{x}_t \in dx | N_{0,t}) = \frac{\bar{q}_t(x)dx}{\int_{\mathbb{R}^d} \bar{q}_t(\xi)d\xi}, \quad (5)$$

and also define $\bar{v}_t(x) := \mathbb{E}^\dagger[\Lambda_{t,T}|N_{0,T}]$. Then, it can be shown (Elliott & Malcolm (2005)) that for any measurable function $f$,

$$\mathbb{E}[f(\boldsymbol{x}_t)|N_{0,T}] = \frac{\mathbb{E}^\dagger[\Lambda_{0,T}f(\boldsymbol{x}_t)|N_{0,T}]}{\mathbb{E}^\dagger[\Lambda_{0,T}|N_{0,T}]} = \frac{\int_{\mathbb{R}^d} f(\xi)\bar{q}_t(\xi)\bar{v}_t(\xi)d\xi}{\int_{\mathbb{R}^d} \bar{q}_t(\xi)\bar{v}_t(\xi)d\xi}. \tag{6}$$

Next, recalling that $h(\cdot)$ is the mapping to ensure the intensity process is positive, define the function $\Psi_t$ for a given sample path of count observations (i.e., *pathwise*) as

$$\Psi_t := \Psi(h(x), t, N_t) = \exp\left[(1 - h(x))t + N_t \log h(x)\right], \forall x \in \mathbb{R}^d.$$

Following Elliott & Malcolm (2005, Theorem 4) one may use $\Psi_t$ to derive a coupled system of pathwise SPDEs that characterize $\bar{q}_t(x)$ and $\bar{v}_t(x)$. In particular, they show that $q_t = \Psi_t^{-1}\bar{q}_t$ is a solution to the following SPDE

$$\partial_t q_t(x) = \Psi_t^{-1} L^*[\Psi_t q_t(x)], \forall t \leq T, q_0(x) = \delta_{x_0}(x), \tag{7}$$

where $L^*$ is the adjoint of $L[F(x)] = \frac{1}{2}\sum_{i,j} a_{i,j}(x)\partial_{x_i x_j}F(x) + \sum_i b_i(x)\partial_{x_i}F(x)$, which is the infinitesimal generator of the prior process for any twice-differentiable, continuous, and bounded function $F : \mathbb{R}^d \mapsto \mathbb{R}$ and $a(x) = \sigma(x)\sigma(x)^T$, and $\delta_{x_0}(x)$ is the Dirac delta distribution at $x_0$. Moreover, they also show that $v_t(x) = \Psi_t \bar{v}_t(x)$ satisfies the following backward parabolic equation

$$\partial_t v_t(x) = -\Psi_t L[\Psi_t^{-1} v_t(x)], \tag{8}$$

with terminal condition $v_T(x) = \Psi_T(x)$.

Now it follows from equation 6 that using the solution of these two SPDEs, the marginal smoothing posterior density for any $t \in [0, T]$ satisfies

$$p_t(x|N_{0,T}) = \frac{q_t(\xi)v_t(\xi)d\xi}{\int_{\mathbb{R}^d} q_t(\xi)v_t(\xi)d\xi}. \tag{9}$$

Using the SPDEs in equation 7, and 8, together with 9, it can be shown that the marginal smoothing posterior density $p_t(x|N_{0,T})$ satisfies its own SPDE: for any $t \in [0, T]$,

$$\partial_t p_t(x|N_{0,T}) = -\sum_i \partial_{x_i}\left[\left\{(a(x)[\nabla \log(\Psi_t^{-1}v_t(x))])_i + b_i(x)\right\} p_t(x|N_{0,T})\right]$$
$$+ \frac{1}{2}\sum_{i,j} \partial_{x_i x_j} a_{i,j}(x)p_t(x|N_{0,T}) \tag{10}$$

and $p_t(x|N_{0,T}) = \delta_{x_0}(x)$ with $x_0 = \mathbf{0}$. We present a detailed derivation in Appendix A.1. Corresponding to this SPDE, there exists a *smoothing posterior SDE*, defined as

$$d\bar{\boldsymbol{x}}_t = \left\{a(\bar{\boldsymbol{x}}_t)[\nabla \log(\Psi_t^{-1}v_t(\bar{\boldsymbol{x}}_t))] + b(\bar{\boldsymbol{x}}_t)\right\} dt + \sigma(\bar{\boldsymbol{x}}_t)d\bar{\mathbf{B}}_t \text{ and } \bar{\boldsymbol{x}}_0 = \mathbf{0}, \tag{11}$$

where $\{\bar{\boldsymbol{x}}_t\}$ is a modification of the process $\{\boldsymbol{x}_t\}$ such that $\bar{\mathbf{B}}_t$ is independent of the Cox process $\mathbf{N}_t$ (and thus $\mathbf{B}_t$).

Observe that the entire sample path of the count observations $N_{0,T}$ is summarized through the pathwise function $\Psi_t$ and $v_t$ together in the drift term of this SDE. Also note that the diffusion coefficient of the smoothing posterior SDE is precisely the same as that of the prior SDE. The computation of the drift term in the smoothing posterior SDE requires the solving equation 8 for $v_t(x)$ which, in turn, makes the posterior computation challenging and computationally intractable in general. Consequently, the computation of the marginal posterior density (and hence the path measure). Therefore, we propose a variational inference-based method to compute an approximation to the solution of the smoothing posterior SPDE, by computing an approximate solution to the smoothing posterior SDE in equation 11.

## 3 VARIATIONAL BAYES FOR APPROXIMATING THE SMOOTHING DENSITY

Observe that the posterior path measure is the minimizer of the following variational optimization problem (Mitter & Newton (2003, Proposition 2.1)),

$$\max_{\Pi(\boldsymbol{x}_{0,T})\in\mathcal{P}(\mathcal{C})} \left\{\text{KL}(\Pi\|\Pi_0) + \int d\Pi(\boldsymbol{x}_{0,T}) \log \mathcal{L}(N_{0,T}|h(\boldsymbol{x}_{0,T}))\right\}, \tag{12}$$

where $\mathcal{P}(\mathcal{C})$ is the space of all absolutely continuous measures with respect to $\Pi_0$, the measure induced by a solution of the intensity SDE (equation 2) on the space $\mathcal{C}[0, T]$ of continuous functions with support $[0, T]$, and KL denotes the Kullback-Leibler divergence between two absolutely continuous measures. Note that $\mathcal{P}(\mathcal{C})$ also contains $\Pi_0$.

Solving this optimization problem over all measures in $\mathcal{P}(\mathcal{C})$ is intractable. Therefore, we choose the subset of absolutely continuous measures $\mathcal{Q}_{\bar{b}} \subset \mathcal{P}(\mathcal{C})$ induced by solutions of the following SDE:

$$d\boldsymbol{x}_t = \bar{b}(\boldsymbol{x}_t, N_t, t)dt + \sigma(\boldsymbol{x}_t)d\mathbf{B}_t, \text{ for } t \leq T \text{ and } \boldsymbol{x}_0 = \mathbf{0}, \tag{13}$$

where $\bar{b}(\cdot, \cdot, \cdot) : \mathbb{R}^d \times \mathbb{N} \times [0, T] \mapsto \mathbb{R}^d$ is the drift function. We term this space of measures $\mathcal{Q}_{\bar{b}}$ as the *variational family*. The measures in $\mathcal{Q}_{\bar{b}}$ are absolutely continuous with respect to $\Pi_0$ as they have the same diffusion coefficient, therefore $\mathcal{Q}_{\bar{b}} \subset \mathcal{P}(\mathcal{C})$. This choice of the variational family is not arbitrary, but rather motivated by the smoothing posterior SDE derived in equation 11, where the diffusion coefficient is $\sigma(\cdot)$ and the drift coefficient has an intractable form (that depends on the prior drift $b(\cdot)$ and diffusion coefficient $\sigma(\cdot)$, and $N_t$ through $\Psi_t$ and $v_t(\cdot)$). Notice that the choice of drift function spans the space of measures in the variational family $\mathcal{Q}_{\bar{b}}$.

Since, $\mathcal{Q}_{\bar{b}} \subset \mathcal{P}(\mathcal{C})$, it follows from equation 12 that

$$\max_{\Pi(\boldsymbol{x}_{0,T}) \in \mathcal{P}(\mathcal{C})} \left\{ -\text{KL}(\Pi\|\Pi_0) + \int d\Pi(\boldsymbol{x}_{0,T}) \log \mathcal{L}(N_{0,T}|h(\boldsymbol{x}_{0,T})) \right\},$$

$$\geq \max_{Q \in \mathcal{Q}_{\bar{b}}} \left\{ -\text{KL}(Q\|\Pi_0) + \int dQ(\boldsymbol{x}_{0,T}) \log \mathcal{L}(N_{0,T}|h(\boldsymbol{x}_{0,T})) \right\} \tag{14}$$

The right hand side above is known as the *evidence lower bound* (ELBO). The corresponding ELBO maximization problem to compute the optimal $Q \in \mathcal{Q}_{\bar{b}}$ (for a given sample path $N_{0,T}$) is simply

$$Q^*(\cdot|N_{0,T}) = \arg\max_{Q \in \mathcal{Q}_{\bar{b}}} \mathbb{E}_Q \left\{ \log \mathcal{L}(N_{0,T}|\boldsymbol{x}_{0,T}) \right\} - \text{KL}(Q\|\Pi_0). \tag{15}$$

Note that absolutely continuous measures on path space correspond to changes in the drift function, for a fixed diffusion coefficient (else, the measures are singular). As a consequence of Girsanov's theorem (Oksendal, 2013, Theorem 8.6.8) (see Appendix A.2 for the proof) we have,

$$\text{KL}(Q\|\Pi_0) = \frac{1}{2}\mathbb{E}_Q \left[ \int_0^T \|\sigma^{-1}(\boldsymbol{x}_t)(b(\boldsymbol{x}_t) - \bar{b}(\boldsymbol{x}_t, N_t, t))\|^2 dt \right], \tag{16}$$

where recall $b(\cdot, \cdot)$ is the drift of the prior SDE defined in equation 2 and $\bar{b}(\cdot, \cdot, \cdot)$ is the drift of the variational SDE. Substituting this into equation 15 yields

$$Q^*(\cdot|N_{0,T}) = \arg\max_{Q \in \mathcal{Q}_{\bar{b}}} \mathbb{E}_Q \left\{ \log \mathcal{L}(N_{0,T}|\boldsymbol{x}_{0,T}) - \frac{1}{2} \int_0^T \|\sigma^{-1}(\boldsymbol{x}_t)[b(\boldsymbol{x}_t, t) - \bar{b}(\boldsymbol{x}_t, N_t, t)]\|^2 dt \right\}. \tag{17}$$

We denote $Q^*(\cdot|N_{0,T})$ as the *variational Bayesian smoothing posterior (VBSP) path measure* . Next, we lay down the details of the SVI algorithm to solve the above optimization problem to compute the VBSP.

## 4 STOCHASTIC VARIATIONAL INFERENCE OF THE VBSP

It is evident from the ELBO in equation 17 and the definition of the variational family $\mathcal{Q}_{\bar{b}}$ that computing the VBSP measure entails the computation of the unknown drift function $\bar{b}(\cdot, \cdot, \cdot)$ in equation 13. We further restrict the family of measures $\mathcal{Q}_{\bar{b}}$ by assuming the drift functions belong to a class of parametrized, smooth functions. A feasible way to model this class of drift functions is through a neural network. However, it is possible to use simpler approximation function classes as done in, for example, Sutter et al. (2016), who fixed $\bar{b}$ to ensure that the marginal distributions of the variational smoothing SDE belong to a specific exponential family of distributions. We note

that in choosing a parametrized class, we must still ensure that the resulting drift functions are Lipschitz continuous and satisfy sufficient regularity so that a solution to the SDE equation 13 exists. Furthermore, restricting the drift functions in this way entails a further restriction of the class of (approximating) path measures. An open question is here is how much of a loss this entails (in terms of the Kullback-Leibler divergence from the 'true' posterior path measure in this instance).

To fix the idea, we assume that $\bar{b}(\cdot, \cdot, \cdot)$ lies in a general class of functions parametrized by $\theta$. Henceforth, we write the drift coefficient as $\bar{b}(\cdot, \cdot, \cdot, \theta)$ to make its dependence on $\theta$ explicit. We use stochastic gradient descent (SGD) to maximize the ELBO, which requires the computation of stochastic gradients of the ELBO with respect to $\theta$. To compute the gradients, we generate sample paths of $\boldsymbol{x}^{\theta}$, the solution of the variational SDE equation 13 for a given $\theta$, using a first-order Euler-Maruyama integration of the SDE. We do this for convenience, though higher order approximations could be used. Specifically, we partition the time interval $[0, T]$ in $K$ equal sub-intervals of length $\Delta = T/K$, $\{t_0, t_1, \ldots t_K\}$, where $t_0 = 0$ and $t_K = T$ and then generate the sequence of $\{\boldsymbol{x}_{t_i}^{\theta}\}_{i=1}^{K}$ using the following recursive equation and initial condition $\boldsymbol{x}_{t_0}^{\theta} = 0$:

$$\boldsymbol{x}_{t_i}^{\theta} = \boldsymbol{x}_{t_{i-1}}^{\theta} + \bar{b}(\boldsymbol{x}_{t_{i-1}}^{\theta}, N_{t_{i-1}}, t_{i-1}, \theta)\Delta + \Delta\sigma(\boldsymbol{x}_{i-1}^{\theta})Z_i, \forall \ i \in \{1, 2, \ldots K\} \tag{18}$$

where $\{Z_i\}_{i=1}^{K}$ is the sequence of $K$ independent and identically distributed (i.i.d.) $d-$dimensional standard Gaussian random vectors.

We generate $M$ independent sample paths of the discrete-time process in equation 18 denoted as $\{\boldsymbol{x}_{t_i}^{\theta,m}\}$, for $m \in \{1, 2, \ldots M\}$, to compute a sample average approximation (SAA) of the ELBO over the partition $\{t_0, t_1, \ldots t_K\}$ as

$$\widehat{\text{ELBO}} = \frac{1}{M} \sum_{m=1}^{M} \left[ \sum_{i=0}^{K-1} \left[ \log \mathcal{L}(N_{t_i, t_{i+1}} | h(\boldsymbol{x}_{t_i}^{\theta,m})\Delta) - \frac{1}{2}\|\sigma(\boldsymbol{x}_{t_i}^{\theta,m})^{-1}[b(\boldsymbol{x}_{t_i}^{\theta,m}, t_i) - \bar{b}(\boldsymbol{x}_{t_i}^{\theta,m}, N_{t_i}, t_i, \theta)]\|^2\Delta \right] \right]$$

$$= \frac{1}{M} \sum_{m=1}^{M} \left[ \sum_{i=0}^{K-1} \left[ N_{t_i, t_{i+1}} \log(h(\boldsymbol{x}_{t_i}^{\theta,m})\Delta) - h(\boldsymbol{x}_{t_i}^{\theta,m})\Delta - \frac{1}{2}\|\sigma(\boldsymbol{x}_{t_i}^{\theta,m})^{-1}[b(\boldsymbol{x}_{t_i}^{\theta,m}, t_i) - \bar{b}(\boldsymbol{x}_{t_i}^{\theta,m}, N_{t_i}, t_i, \theta)]\|^2\Delta \right] \right] + \text{C}, \tag{19}$$

where we used the definition of the Poisson likelihood $\mathcal{L}$ from equation 1 and C is a constant independent of $\theta$. Now, to compute gradients of ELBO with respect to $\theta$, observe that the gradient operator can be exchanged with the expectation, since the only source of randomness in each sample path of $\boldsymbol{x}_t^{\theta,m}$ are $K$ i.i.d. Gaussian random vectors $\{Z_i^m\}_{i=1}^{K}$, which are independent of $\theta$. Notice too that this is a pathwise analog of the reparametrization trick, and has been used recently to learn deep latent models (Tzen & Raginsky, 2019; Li et al., 2020). Also, note that thus far the $\widehat{\text{ELBO}}$ is defined for a single sample path of the count observations $N_{0,T}$. However, as noted before, in our method we will also take a sample average of $\widehat{\text{ELBO}}$ over multiple batches of sample paths of count observations at each epoch of the training algorithm.

## 5 NUMERICAL EXPERIMENTS

We present three experiments demonstrating the efficacy and utility of our proposed SVI method. First, we consider a setting when the underlying stochastic intensity process is 1-dimensional. We compare the SVI approximation with the 'true' smoothing posterior density computed using the solution of the forward and backward SPDEs, defined in equation 7 and equation 8. We solve the SPDEs using a standard finite element method (FEM) solver (Skeel & Berzins, 1990, Matlab solvers for 1-D PDEs). In a second experiment, we demonstrate the performance of our algorithm on a subset of a Bike-sharing dataset obtained from the UCI machine learning repository (Fanaee-T & Gama, 2013). In this experiment we estimate a smoothing posterior density for the observed counts of the demand for bikes in a 24 hour period, assuming that the demand process is well-modeled by a Cox process. In our third and final experiment, we apply our method to compute an approximation to a 4-dimensional smoothing posterior density. We note that despite being low dimensional, the standard FEM solver does not scale to this setting, while our method can be straightforwardly adapted.

## 5.1 Variational approximation of univariate Smoothing posterior density

As defined in Section 2, we are interested in learning the posterior measure over an unknown process $\{\boldsymbol{x}_t \in \mathbb{R}\}$ where the intensity process $\{\boldsymbol{z}_t\}$ satisfies $\boldsymbol{z}_t = h(\boldsymbol{x}_t)$. We set $b(x) = -x$ and $\sigma(x) = 1$ in the prior SDE as defined in equation 2. Furthermore, motivated from the mathematical structure of the true smoothing SDE equation 11, we fix our variational family $\mathcal{Q}$ to be the class of measures induced by solutions to the class of SDEs in equation 13 with drift and diffusion coefficient set as

$$\bar{b}(x, t, N_t) = -\frac{\Psi_t'}{\Psi_t} + V(x, t, N_T - N_t; \theta) - x, \quad \sigma(x) = 1,$$

where $\Psi_t'$ is the derivative of $\Psi_t$ with respect to $x$. Here $V(x, t, N_T - N_t; \theta)$ is modeled using a neural network with 2 hidden layers whose parameters we call $\theta$ (see Appendix A.3 for more details on the architecture).

### 5.1.1 Simulated dataset

We generate sample paths of the count observation $\mathbf{N}_{0,T}$ from a non-homogeneous Poisson process, where the intensity process $\{z_t^0\}$ is the solution of the following ordinary differential equation, $\frac{dz_t^0}{dt} = 20(2 - t)\exp(-0.85(2 - t)^2)$, and $z_0^0 = 0$. We fix the map $h(a) = 5 * \exp(-.08 * (a - 5)^2)$ in this experiment. To train the neural network $V$, we use 150 samples paths of the count observation between time interval $[0, 2]$ and optimize $\widehat{\text{ELBO}}$ in equation 19 using Adam (Kingma & Ba, 2014).

To demonstrate the efficacy of our approach, we first generate 20 test sample paths of count observations and compute the true smoothing posterior density (defined in equation 9) using the solution of the forward and backward SPDEs (defined in equation 7 and equation 8 respectively). We do this using the FEM method. Then for the same test observations we compute the VBSP density by FEM using the trained drift coefficient of VBSP SDE (see equation 10 and 11).

Comparative results are presented in Figure 1. We clearly see from the the top row plot that these are very similar to each other, with our variational approximation capturing the sharp rises in density with high fidelity. Note from the first two plots in the bottom row that as the ELBO decreases the gap between the the true and VB smoothing posterior reduces too. Moreover, the time required to compute the smoothing density using the forward and backward SPDEs on the test data is about 2.2 seconds which is approximately fifty percent higher than the trained VBSP density, which required 1.6 seconds (on an 3.1 GHz Intel i5 CPU). This is due to the fact that we are required to solve one SPDE in the latter case instead of two in the former case.

Notice that the learnt drift of the smoothing SDE equation 13 is a map which can be used with any sample path of count observations to generate Monte Carlo samples from the an approximate smoothing posterior density. In contrast, it is challenging to sample from the true smoothing SDE as it involves computing the solution of the system of SPDEs in equation 7 and 8. Furthermore, this solution must be recomputed for each new sample path of count observations.

### 5.1.2 Bike Sharing Dataset

In this experiment, we compute the VBSP density for the hourly counts of demand in a bike-sharing system. The experimental setting remains unaltered, albeit the diffusion coefficient is set to $\sigma(x) = 1.1$, to capture the increased variability in counts of bike demand than the variability in simulated counts in the previous experiment. Notice that fixing the map $h$ is a modeling question, and we consider mappings of the form $h(x) = a\exp(-b(x - c)^2)$ parametrized by $a, b,$ and $c$. We take a simple empirical Bayes heuristic to fix their values after observing the count observations, based on the fact that $h(x)\Delta$ is the mean of the Poisson counts in the interval $\Delta$. We set $a$ in such a way that $h(x)\Delta$ equals the maximum of the median observed count. For the Bike-sharing data we re-scaled the problem to interval $[0, 2]$ and fixed $\Delta = 0.083$ and thus choose $90/0.083 \simeq 1050$ as $a$. The choice of $b$ and $c$ depends on the diffusion coefficient $\sigma(x)$ and $x_0$, as the SDE should be able to explore the relevant domain of $h$ to appropriately model the actual count observations. Thus, after looking at the count data, we chose $h(x) = 1050 * \exp(-0.001 * (y - 50)^2)$.

The empirical results are summarized in Figure 2. We note here that the FEM approach (our implementation) to compute the VBSP density was numerically unstable and failed; this may be attributed

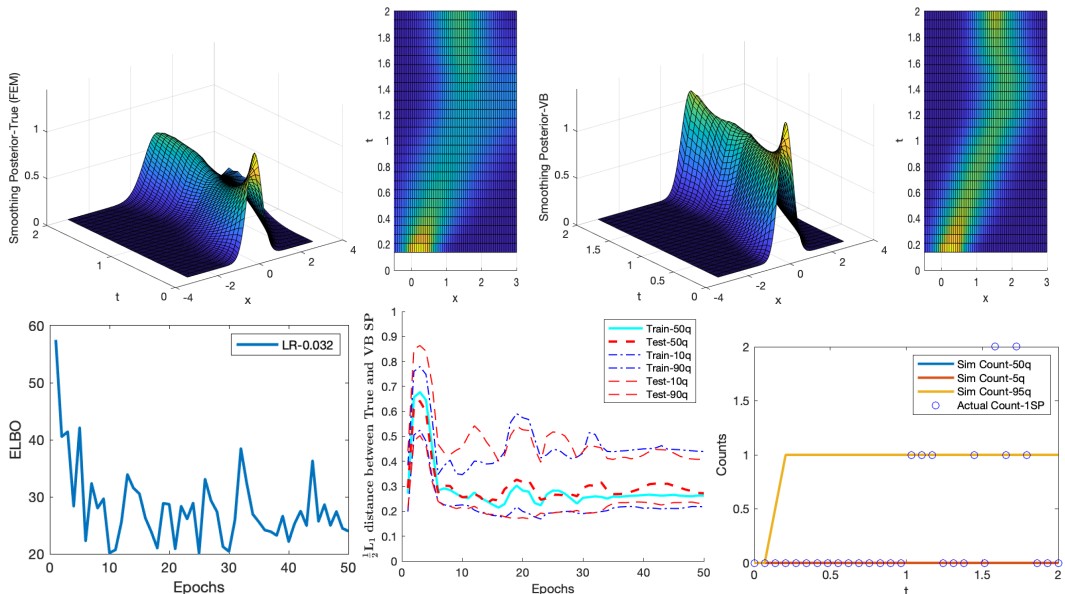

Figure 1: Variational vs. True smoothing posterior density on the 1-D simulated dataset. The top row compares the variational approximation with the true smoothing density. The bottom-left plot shows the ELBO as a function of epochs of the training algorithm. We also compute the $L_1$ distance between the VB and true smoothing density over 20 sample paths of both training and test count observations as the training progresses and plot the $10^{th}$, $50^{th}$ and $90^{th}$ quantile in the bottom-middle figure. The bottom-right plot depicts that 95% of the test count observations are within the $5^{th}$ and $95^{th}$ quantile of the simulated counts, where counts are simulated using the learned VBSP SDE for that test sample path of count observations.

to the concentrated nature of the VBSP, as visible in the plots. It is evident from the last plot in Figure 2 that for the current $h$, VBSP density captures the trend in the real count observations well, however we anticipate that a different choice of the map $h$ can better model the count observations.

Note that in a smoothing problem, the prior intensity process (specifically the drift coefficient $b(\cdot)$, diffusion coefficient $\sigma(\cdot)$ and the map $h(\cdot)$) are sourced from an expert, and the objective is to update the modeler's beliefs with count observations to compute the smoothing posterior density. In many settings, the functions $b(\cdot), \sigma(\cdot)$ and $h(\cdot)$ are known only up to some unknown parameters. It is fairly straightforward to combine the parameters of $h$ and $b$ with the neural network parameters $\theta$, and learn them all in a data-driven manner. We choose not to do this to keep the discussion simple. Learning the parameters of $\sigma$ presents a slightly greater challenge since the path measures for different settings of $\sigma$ are singular. This is a crucial difference from the finite dimensional setting, where the Lebesgue measure is a common reference. We leave solving this problem for future work.

## 5.2 VARIATIONAL APPROXIMATION OF MULTIVARIATE SMOOTHING POSTERIOR DENSITY

We demonstrate our method on a 4-dimensional smoothing problem. In this case, we fix the map $h(\boldsymbol{a}) = 25 * \exp(-.08 * \|\boldsymbol{a} - 5\|^2)$, where $\| \cdot \|$ is the L-2 norm, and $\boldsymbol{a} \in \mathbb{R}^4$. We also choose the prior density to be induced by an SDE defined in equation 2 with $b(\boldsymbol{x}) = -\boldsymbol{x}$ and $\sigma(\boldsymbol{x}) = \boldsymbol{I}$, where $\boldsymbol{x} \in \mathbb{R}^d$ and $\boldsymbol{I}$ is a $d \times d$ identity matrix. Furthermore, we choose our variational family to be a family of SDE as defined in equation 13, with drift and diffusion coefficients, $\bar{b}(\boldsymbol{x}, t, N_t) = -\frac{\nabla \Psi_t}{\Psi_t} + V_d(\boldsymbol{x}, t, N_T - N_t; \theta) - \boldsymbol{x}$ and $\sigma(\boldsymbol{x}) = \boldsymbol{I}$.

To train the neural network $V$, we use 150 samples paths of the count observation between time interval $[0, 2]$ and optimize the ELBO defined in equation 19 using Adam (Kingma & Ba, 2014). We plot the empirical results in Figure 3.

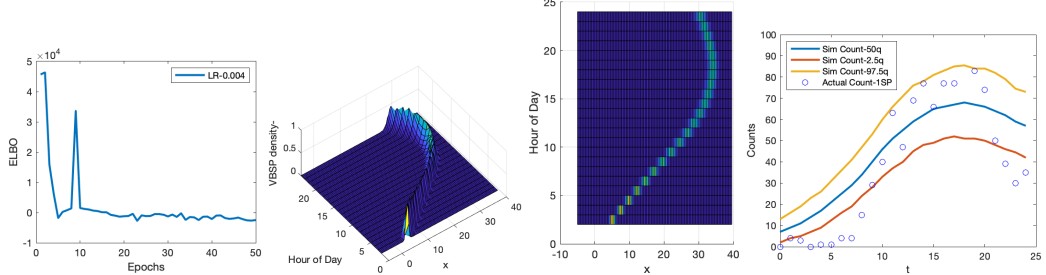

Figure 2: VBSP smoothing posterior density for Bike Sharing Data. The left plot shows the ELBO as the training progresses. The second and third plots depict the empirical VBSP density computed on a test sample path of count observations, using 1000 simulated sample paths of the learnt variational smoothing SDE. In the right plot, we use the same test sample path of count observation, compute the VBSP using the trained drift and demonstrate that most count observations (of the test sample path) ( 70%) lie within the $97.5^{th}$ and $2.5^{th}$ quantile of the simulated counts.

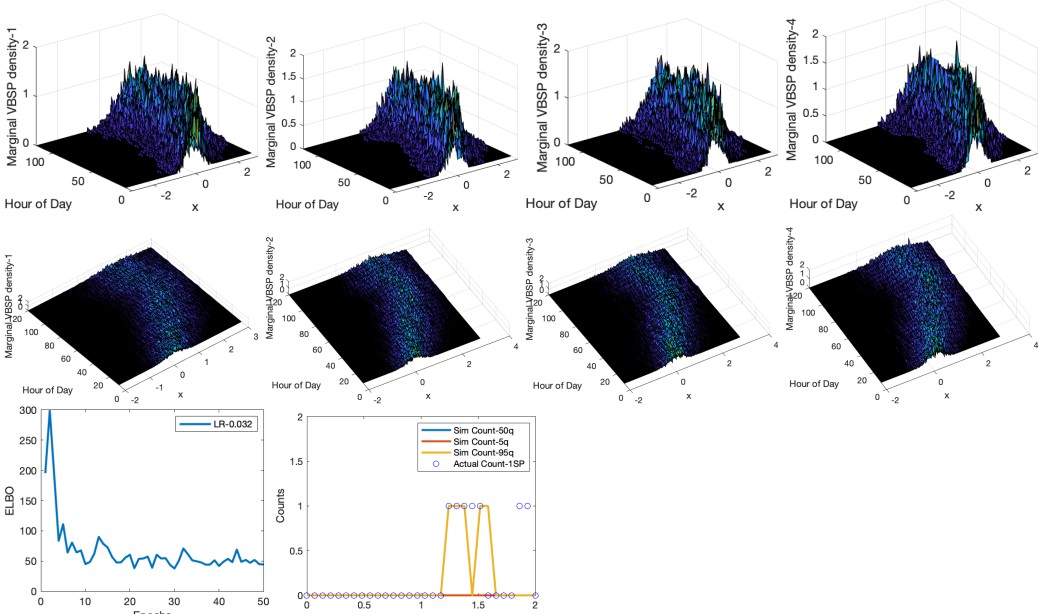

Figure 3: Multivariate VB smoothing posterior density: The first two rows of Figure 3 show the marginals of the (empirical) VBSP density using the trained drift coefficient for a given test sample path of count observations. The bottom-left figure plots the ELBO value as a function of the number of training epochs. Then, for a test sample path of count observations, the right hand plot shows that more than 95% of these observations are within the 95% confidence interval of the simulated count observations generated from the trained 4-dimensional VBSP density computed for that test path.

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

# A    APPENDIX

## A.1    DERIVATION OF SMOOTHING SPDE

According to Theorem * [sic] in Elliott & Malcolm (2005), $K_t := (\int_{\mathbb{R}^d} q_t(\xi) v_t(\xi) d\xi)^{-1}$ is almost surely constant for $t \le T$. Using this result it follows that

$$
\begin{aligned}
\partial_t p_{S,t} &= K_t q_t(x) \partial_t v_t(x) + K_t v_t(x) \partial_t q_t(x) \\
&= K_t q_t(x) \partial_t v_t(x) + K_t v_t(x) \left[ \Psi_t^{-1} L^* [\Psi_t \frac{p_{S,t}}{K_t v_t(x)}] \right] \\
&= -\frac{p_{S,t}}{v_t(x)} \Psi_t L[\Psi_t^{-1} v_t(x)] + v_t(x) \left[ \Psi_t^{-1} L^* [\Psi_t \frac{p_{S,t}}{v_t(x)}] \right] \\
&= -\frac{p_{S,t}}{V_t(x)} L[V_t(x)] + V_t(x) \left[ L^* [\frac{p_{S,t}}{V_t(x)}] \right].
\end{aligned}
\tag{20}
$$

where $V_t(x) = \Psi_t^{-1} v_t(x)$ and $P_{S,t} = p_t(x|N_{0,T})$ are introduced for brevity. Now observe that

$$
\begin{aligned}
V_t(x) L^* \left[ \frac{p_{S,t}}{V_t(x)} \right] &= V_t(x) \left[ \frac{1}{2} \sum_{i,j} \partial_{x_i x_j} \left( a_{i,j}(x) \frac{p_{S,t}}{V_t(x)} \right) - \sum_i \partial_{x_i} \left( b_i(x) \frac{p_{S,t}}{V_t(x)} \right) \right] \\
&= V_t(x) \frac{1}{2} \sum_{i,j} \partial_{x_i x_j} \left( a_{i,j}(x) \frac{p_{S,t}}{V_t(x)} \right) + \sum_i \left[ \frac{\partial_{x_i} V_t(x) b_i(x) p_{S,t}}{V_t(x)} \right] \\
&\quad - \sum_i \left[ (\partial_{x_i} b_i(x) p_{S,t} + b_i(x) \partial_{x_i} p_{S,t}) \right].
\end{aligned}
\tag{21}
$$

Consider the summand in the first term of equation 21 and observe that

$$
\begin{aligned}
\partial_{x_i x_j} \left[ \frac{a_{i,j}(x) p_{S,t}}{V_t(x)} \right] &= \partial_{x_i} \left[ \frac{V_t(x) \partial_{x_j} (a_{i,j}(x) p_{S,t}) - \partial_{x_j} V_t(x) a_{i,j}(x) p_{S,t}}{V_t^2(x)} \right] \\
&= \frac{V_t(x) \partial_{x_i x_j} (a_{i,j}(x) p_{S,t}) - \partial_{x_i} V_t(x) \partial_{x_j} (a_{i,j}(x) p_{S,t})}{V_t^2(x)} \\
&\quad + \frac{2 \partial_{x_i} V_t(x)}{V_t^3(x)} \left[ \partial_{x_j} V_t(x) (a_{i,j}(x) p_{S,t}) \right] - \frac{1}{V_t^2(x)} \partial_{x_i} \left( \partial_{x_j} V_t(x) a_{i,j}(x) p_{S,t} \right)
\end{aligned}
$$

Now the second term in equation 21 can be expressed as

$$
\begin{aligned}
V_t(x) \partial_{x_i x_j} \left[ \frac{a_{i,j}(x) p_{S,t}}{V_t(x)} \right] &= \partial_{x_i x_j} (a_{i,j}(x) p_{S,t}) - \frac{\partial_{x_i} V_t(x) \partial_{x_j} (a_{i,j}(x) p_{S,t})}{V_t(x)} \\
&\quad + \frac{2 \partial_{x_i} V_t(x)}{V_t^2(x)} \left[ \partial_{x_j} V_t(x) (a_{i,j}(x) p_{S,t}) \right] - \frac{1}{V_t(x)} \partial_{x_i} \left[ \partial_{x_j} V_t(x) a_{i,j}(x) p_{S,t} \right]
\end{aligned}
\tag{22}
$$

Substituting the above expression in equation 21, we have

$$
\begin{aligned}
V_t(x) L^* \left[ \frac{p_{S,t}}{V_t(x)} \right] &= \frac{1}{2} \sum_{i,j} \left\{ \partial_{x_i x_j} (a_{i,j}(x) p_{S,t}) - \frac{\partial_{x_i} V_t(x) \partial_{x_j} (a_{i,j}(x) p_{S,t})}{V_t(x)} \right. \\
&\quad \left. + \frac{2 \partial_{x_i} V_t(x)}{V_t^2(x)} \left[ \partial_{x_j} V_t(x) (a_{i,j}(x) p_{S,t}) \right] - \frac{1}{V_t(x)} \partial_{x_i} \left[ \partial_{x_j} V_t(x) a_{i,j}(x) p_{S,t} \right] \right\} \\
&\quad + \sum_i \left[ \frac{\partial_{x_i} V_t(x) b_i(x) p_{S,t}}{V_t(x)} \right] - \sum_i \left[ (\partial_{x_i} b_i(x) p_{S,t} + b_i(x) \partial_{x_i} p_{S,t}) \right].
\end{aligned}
\tag{23}
$$

Now consider the first term in the RHS of equation 20

$$
-\frac{p_{S,t}}{V_t(x)} L[V_t(x)] = -\frac{1}{2} \sum_{i,j} \frac{a_{i,j}(x) p_{S,t}}{V_t(x)} \partial_{x_i x_j} V_t(x) - \sum_i \frac{b_i(x) p_{S,t}}{V_t(x)} \partial_{x_i} V_t(x).
\tag{24}
$$

Now substituting equation 23 and equation 24 into equation 20, we obtain

$$
\begin{aligned}
\frac{\partial}{\partial t} p_{S,t} &= -\frac{p_{S,t}}{V_t(x)} L[V_t(x)] + V_t(x)\left[L^*\left[\frac{p_{S,t}}{V_t(x)}\right]\right] \\
&= -\frac{1}{2}\sum_{i,j}\frac{a_{i,j}(x)p_{S,t}}{V_t(x)}\partial_{x_i x_j}V_t(x) - \sum_i \frac{b_i(x)p_{S,t}}{V_t(x)}\partial_{x_i}V_t(x) + \frac{1}{2}\sum_{i,j}\Big\{\partial_{x_i x_j}\left(a_{i,j}(x)p_{S,t}\right) \\
&\quad - \frac{\partial_{x_i}V_t(x)\partial_{x_j}\left(a_{i,j}(x)p_{S,t}\right)}{V_t(x)} + \frac{2\partial_{x_i}V_t(x)}{V_t^2(x)}\left[\partial_{x_j}V_t(x)\left(a_{i,j}(x)p_{S,t}\right)\right] - \frac{1}{V_t(x)}\partial_{x_i}\left[\partial_{x_j}V_t(x)a_{i,j}(x)p_{S,t}\right]\Big\} \\
&\quad + \sum_i\left[\frac{\partial_{x_i}V_t(x)b_i(x)p_{S,t}}{V_t(x)}\right] - \sum_i\left[(\partial_{x_i}b_i(x)p_{S,t} + b_i(x)\partial_{x_i}p_{S,t})\right] \\
&= \frac{1}{2}\sum_{i,j}\partial_{x_i x_j}\left(a_{i,j}(x)p_{S,t}\right) - \sum_i\partial_{x_i}\left[b_i(x)p_{S,t}\right] - \frac{1}{2}\sum_{i,j}\Big\{\frac{\partial_{x_i x_j}V_t(x)}{V_t(x)}[a_{i,j}(x)p_{S,t}] \\
&\quad + \frac{\partial_{x_i}V_t(x)\partial_{x_j}\left(a_{i,j}(x)p_{S,t}\right)}{V_t(x)} - \frac{2\partial_{x_i}V_t(x)\partial_{x_j}V_t(x)}{V_t^2(x)}[(a_{i,j}(x)p_{S,t})] \\
&\quad + \frac{1}{V_t(x)}\left[\partial_{x_j}V_t(x)\partial_{x_i}(a_{i,j}(x)p_{S,t}) + \partial_{x_i x_j}V_t(x)a_{i,j}(x)p_{S,t}\right]\Big\} \\
&= \frac{1}{2}\sum_{i,j}\partial_{x_i x_j}\left(a_{i,j}(x)p_{S,t}\right) - \sum_i\partial_{x_i}\left[b_i(x)p_{S,t}\right] \\
&\quad - \frac{1}{2}\sum_{i,j}\Big\{\partial_{x_i}\left[\partial_{x_j}\log V_t(x)[a_{i,j}(x)p_{S,t}]\right] + \frac{\partial_{x_i}V_t(x)\partial_{x_j}\left(a_{i,j}(x)p_{S,t}\right)}{V_t(x)} \\
&\quad - \frac{\partial_{x_i}V_t(x)\partial_{x_j}V_t(x)}{V_t^2(x)}[(a_{i,j}(x)p_{S,t})] + \frac{1}{V_t(x)}\left[\partial_{x_i x_j}V_t(x)a_{i,j}(x)p_{S,t}\right]\Big\}
\end{aligned}
$$

Since, $\partial_{x_i x_j}V_t(x) = \partial_{x_j x_i}V_t(x)$ therefore

$$
\begin{aligned}
\frac{\partial}{\partial t} p_{S,t} &= \frac{1}{2}\sum_{i,j}\partial_{x_i x_j}\left(a_{i,j}(x)p_{S,t}\right) - \sum_i\partial_{x_i}\left[b_i(x)p_{S,t}\right] \\
&\quad - \frac{1}{2}\sum_{i,j}\Big\{\partial_{x_i}\left[\partial_{x_j}\log V_t(x)[a_{i,j}(x)p_{S,t}]\right] + \partial_{x_j}\left[\partial_{x_i}\log V_t(x)[a_{i,j}(x)p_{S,t}]\right]\Big\} \\
&= \frac{1}{2}\sum_{i,j}\partial_{x_i x_j}\left(a_{i,j}(x)p_{S,t}\right) - \sum_i\partial_{x_i}\left[b_i(x)p_{S,t}\right] - \sum_{i,j}\Big\{\partial_{x_i}\left[\partial_{x_j}\log V_t(x)[a_{i,j}(x)p_{S,t}]\right]\Big\} \\
&= \frac{1}{2}\sum_{i,j}\partial_{x_i x_j}\left(a_{i,j}(x)p_{S,t}\right) - \sum_i\partial_{x_i}\left\{[(a(x)[\nabla\log V_t(x)])_i + b_i(x)]p_{S,t}\right\}.
\end{aligned}
\tag{25}
$$

## A.2 KL-DIVERGENCE BETWEEN A MEMBER OF VARIATIONAL FAMILY AND PRIOR SDE

We derive a pathwise expression for $\text{KL}(Q\|\Pi_0)$ for a given count observation path $N_{0,T}$.

**Theorem A.1.** *Define* $u(x_t, t, N_t; \theta) := \sigma(x_t)^{-1}\left(b(x_t, t) - \bar{b}(x_t, t, N_t; \theta)\right)$ *and suppose that* $u$ *satisfies a strong Novikov's condition:*

$$
\mathbb{E}\left[\exp\left(\frac{1}{2}\int_0^T \|u(x_t, t, N_t; \theta)\|^2 dt\right)\right] < +\infty \ \forall \theta, \phi.
$$

*Then,*

$$
\text{KL}(Q\|\Pi_0) = \mathbb{E}_Q\left[\frac{1}{2}\int_0^T \|u(x_t, t, N_t; \theta)\|^2 ds\right].
\tag{26}
$$

*Proof.* Given samples path of count observation $N_{0,T}$, using the definition of $u$ and under Novikov's condition, using Girsanov's theorem (Oksendal, 2013, Theorem 8.6.8), we have

$$\frac{dQ}{d\Pi_0} = \exp\left(-\int_0^t u(x_t, t, N_t; \theta)d\mathbf{B}_s - \frac{1}{2}\int_0^t \|u(x_t, t, N_t; \theta)\|^2 ds\right),$$

and

$$\hat{\mathbf{B}}_t := \int_0^t u(x_t, t, N_t; \theta)ds + \mathbf{B}(t) \tag{27}$$

is a Brownian motion w.r.t. $Q$. Furthermore, we also have

$$dx_t = b(x_t, t)dt + \sigma(x_t)d\hat{\mathbf{B}}_t. \tag{28}$$

The following expression is obtained by substituting for $\frac{dQ}{d\Pi_0}$:

$$\begin{aligned}
&\mathbb{E}_Q\left[\log\left(\frac{dQ(x_{0:T})}{d\Pi_0(x_{0:T})}\right)\right] \\
&= -\mathbb{E}_Q\left[\int_0^T \left(u(x_t, t, N_t; \theta)d\mathbf{B}_s + \frac{1}{2}\|u(x_t, t, N_t; \theta)\|^2\right)ds\right].
\end{aligned} \tag{29}$$

Now, applying $\hat{\mathbf{B}}(t) := \int_0^t u(x_t, t, N_t; \theta)ds + \mathbf{B}_t$ in Equation equation 29, we have

$$\begin{aligned}
\mathbb{E}_Q\left[\int_0^T u(x_t, t, N_t; \theta)d\mathbf{B}_s\right] &= \mathbb{E}_Q\left[\int_0^T u(x_t, t, N_t; \theta)[d\hat{\mathbf{B}}_s - u(x_t, t, N_t; \theta)ds]\right] \\
&= \mathbb{E}_Q\left[\int_0^T u(x_t, t, N_t; \theta)d\hat{\mathbf{B}}_s - \int_0^T \|u(x_t, t, N_t; \theta)\|^2 ds\right] \\
&= \mathbb{E}_Q\left[-\int_0^T \|u(x_t, t, N_t; \theta)\|^2 ds\right].
\end{aligned} \tag{30}$$

Substituting equation 30 into equation 29 yields

$$\begin{aligned}
\mathrm{KL}(Q\|\Pi_0) &= \mathbb{E}_Q\left[\log\left(\frac{dQ(x_{0:T})}{d\Pi_0(x_{0:T})}\right)\right] \\
&= \mathbb{E}_Q\left[\frac{1}{2}\int_0^T \|u(x_t, t, N_t; \theta)\|^2 ds\right].
\end{aligned} \tag{31}$$

and thus concludes the proof. $\square$

### A.3 Neural network architecture

For all the experiments, we use the neural network architecture depicted in Figure A.3, with ReLU activation functions between fully-connected hidden layers.

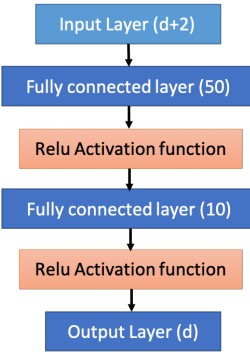

Figure 4: Neural network architecture used in the numerical experiments.

### A.4 Comparing computational time required to numerically compute VBSP and True smoothing density using FEM

| Epochs | 1 | 2 | 3 | 4 | 5 | 6 | 7 | 8 | 9 | 10 | 11 | 12 | 13 | 14 | 15 | 16 | 17 |
|---|---|---|---|---|---|---|---|---|---|---|---|---|---|---|---|---|---|
| VBSP | 1.62 | 1.44 | 1.62 | 1.6 | 1.59 | 1.63 | 1.59 | 1.61 | 1.61 | 1.62 | 1.6 | 1.65 | 1.65 | 1.61 | 1.61 | 1.73 | 1.74 |
| True SD | 2.07 | 2 | 2.22 | 2.17 | 2.16 | 2.16 | 2.23 | 2.12 | 2.12 | 2.15 | 2.12 | 2.17 | 2.14 | 2.11 | 2.14 | 2.18 | 2.15 |
| Epochs | 18 | 19 | 20 | 21 | 22 | 23 | 24 | 25 | 26 | 27 | 28 | 29 | 30 | 31 | 32 | 33 | 34 |
| VBSP | 1.7 | 1.72 | 1.83 | 1.9 | 1.86 | 1.84 | 1.81 | 1.85 | 1.75 | 1.78 | 1.87 | 1.9 | 1.95 | 2.1 | 2.06 | 1.81 | 1.84 |
| True SD | 2.09 | 2.18 | 2.25 | 2.37 | 2.22 | 2.18 | 2.22 | 2.23 | 2.21 | 2.24 | 2.33 | 2.29 | 2.36 | 2.39 | 2.41 | 2.17 | 2.12 |
| Epochs | 35 | 36 | 37 | 38 | 39 | 40 | 41 | 42 | 43 | 44 | 45 | 46 | 47 | 48 | 49 | 50 | |
| VBSP | 1.83 | 1.81 | 1.85 | 1.89 | 1.87 | 1.84 | 1.85 | 1.86 | 1.92 | 1.91 | 1.92 | 1.88 | 1.84 | 1.91 | 1.91 | 1.82 | |
| True SD | 2.14 | 2.14 | 2.15 | 2.18 | 2.25 | 2.17 | 2.14 | 2.12 | 2.2 | 2.15 | 2.23 | 2.17 | 2.17 | 2.28 | 2.25 | 2.17 | |

Table 1: Comparing computational time (in sec) required to compute 1-D VBSP and true smoothing density using FEM. The time reported here are median over 20 test sample paths of count observations at each epoch

