# OpenReview forum: "Variational inference for diffusion modulated Cox processes"
_ICLR.cc/2021/Conference — Reject_

### Official Review · AnonReviewer2 · 2020-10-26
**The paper is clear and well written with original and significant contribution.**

**Rating:** 7
**Confidence:** 3

**Review:**

The paper under review proposes a variational inference procedure for a specific class of Cox processes whose intensity is derived from a stochastic differential equation. The methodology relies on a restriction of candidate solutions the the subset for which the drift depends on $x_t$, $N_t$ and $t$; the drift is then modelled with a neural network. By simulating from the candidate model, a sample average approximation of the ELBO is used to compute a stochastic gradient descent algorithm, optimize the bound and thus estimate non-parametrically the drift.

The paper is clear and well written with original and significant contribution. The paper would benefit from clarifications about the following points:


## Major points
* It is not clear how equation (13) is a subset of the solution of (2) in general more parameters implies more degree of freedom and thus the drift in (13) could look more general than $b(x)$ (the use the $\bar{b}$ notation suggesting it refers more to (11) than (2)). This probably comes from the fact that all variables are indexed by $t$ but more details would be nice here.
* The proposed framework requires to specify both $b$ and $\sigma$ a priori. It would be nice to explain what is the impact and restrictions of doing so. For instance, the first experiment set $\sigma$ to $1$ and the second to $1.1$ without further justification. Could the author(s) elaborate(s) on this point? Similarly, the authors use a very flexible model (NN) for $\bar{b}$, so the main restriction comes from the choice of the `target' $b$: the impact of this choice should be discussed in more detail.
* Similarly to the previous points, in the methodology, the link function $h$ must be set a priori (and its uncertainty is not taken into account). Would it be possible to have an inference methodology estimating not only the drift but also $h$ (and $\sigma$?) at the same time? If not, is it possible to explain why?


## Minor points
* p.1 l.1: the reference for Cox processes is not the 'original' work.
* p.2: 'in particular, we show that...' either 'that' or 'how'.
* p.2: VBSP has never been defined at this stage of the paper.
* p.3: Wiener.
* p.3: the notation $\mathbb{E}^\dagger$ is undefined.
* p.3: defined the indicator function that you are using.
* p.3: shouldn't the $x$ in the definition of $\bar{v}(x_t)$ be bold?
* p.3: which 'section'? Simply 'above'?
* p.3: $f$ denotes first bounded measureable functions and then twice-differentiable ... Use two different notation to avoid confusion.
* In general, use eqref for equations.
* p.5: We call $Q^\star$ as the VB ... Defined the meaning of VB.
* p.5: specify the class of parametric functions used in Sutter et al.
* p.6: the notation $\psi_t'$ is not defined. The link with equation (13) could be made clearer.

---

> ### Author Response · Authors · 2020-11-13
> **Thank you for your comments and positive feedback.**
>
> Please find our response  to your questions/comments below:
>
> ##### Major points
> 1. It is not clear how equation (13) is a .......be nice here. \
> R. Thank you for your comments. You are correct in pointing out that the class of measures induced by (13), denoted as $\mathcal{Q}$$\bar b$, cannot be a subset of the measures induced by the SDE in (2); and we are also not claiming this, as this is false. In the sentence above eq (13), we say that $\mathcal{Q}_{\bar b}$ is a subset of all absolutely continuous measure with respect to $\Pi_0$, which is a larger class of measures that also includes $\Pi_0$. Since the measure induced by SDE (13) is absolutely continuous with respect to $\Pi_0$, therefore it is also included in set $\mathcal{P}(\mathcal{C})$. We have updated the paragraph after eq. (13) to clarify this fact.
> 2. The proposed framework requires to ...... in more detail.\
> R. Thank you for your comments. $b(\cdot)$,  $\sigma(\cdot)$, and $h(\cdot)$ are the parameters of the model. Setting parameters is the task of the modeler, while our focus is mostly on inference given the model. In the interest of clarity, we decided not to include learning these in our paper. Instead, we heuristically set $\sigma(x)$ to $1.1$ in the Bike sharing experiment as the count observations had higher variance than the first experiment as evident from the last plots in Figures 1 and 2 respectively. Moreover, we should point out that the target drift is not $b$, but in fact the target is the intractable drift derived for the smoothing SDE in eq. (11).
> 3. Similarly to the previous points,.......explain why?\
> R. Please refer to our response above.
>
> ##### Minor points
> Thank you for pointing these minor but important errors. We have corrected all of them in the revised paper.
>
> R1. We have added an original reference. (page 1)\
> R2. Removed 'that'. (page 2, para 4, line  2)\
> R3. Defined VBSP (page 2, para 4, line  4)\
> R4. Corrected: 'Wiener' . (line 2 after eq.(2))\
> R5. Definition of $\mathbb{E}^{\dagger}$ added. (line 2 after eq.(3)) \
> R6. We have added the definition of the Indicator function. (line 1 after eq.(4))\
> R7. This is a typo, it should be $\bar v_t(x)$. We have corrected it. (line 1 on page 4) \
> R8. Corrected reference to $h(\cdot)$. (line 1 after eq.(6))\
> R9. We have used two separate notations now $f$ and $F$. (line 1 after eq.(7))\
> R10. We have ensured that eqref is used to refer equations.\
> R11. Defined VB. (line 1 after eq.(17))\
> R12. Specified the class of parametric functions used in Sutter et al. (Sec 4, para 1, line 6)\
> R13. Defined $\Psi_t'$ . (line 5 on page 7)
>
> Note: We have uploaded the revised paper incorporating all your comments and a copy (as supplementary) with changes highlighted in blue.

---

> > ### Comment · AnonReviewer2 · 2020-11-23
> > **Some clarification on Major points**
> >
> > Thanks you very much for taking into account all of my comments. Concerning the the major points:
> >
> > 1. It is now clear maybe at the top of page 5 use 'we choose the subset of absolutely continuous' instead of 'a subset'.
> >
> > 2 and 3. I understand that further parametric/modelling assumptions are necessary depending on the context of the application. However, my comment is not to give detailed explanation on how to make inference of such parameters but to at least explain if this is possible/impossible and then how/why.
> >
> > If that is not possible, then this would be a strong limitation of the methodology because any 'modeller', as you call them, would not be satisfied by 'heuristically setting' $\sigma$ to an arbitrary value which then strongly influence the posterior in (11). Estimating and quantification of such parameters must be possible to ensure that the presented methodology has any kind of practical use and should at least but shortly discussed.

---

> > > ### Author Response · Authors · 2020-11-25
> > > **Thank you for your comment**
> > >
> > > We agree that often, the functions $b(\cdot), \sigma(\cdot)$ and $h(\cdot)$ are known only up to an unknown parameter (say $\theta$) that needs to be learnt in a data-driven manner. Learning the parameters of $h$ and $b$ is relatively straightforward, they can be learnt along with the neural network weights using stochastic gradient descent. Learning the parameters of $\sigma$ presents a slightly greater challenge since the path measures for different settings of $\sigma$ are singular. Learning this is a topic for future research. We have added a paragraph before Sec 5.2 on page 8 in the updated draft to clarify more on this point.

---

> > > > ### Comment · AnonReviewer2 · 2020-11-25
> > > > **Thanks**
> > > >
> > > > This should be good enough. Thanks for taking my remarks into account.
> > > >
> > > > Note: would it really make the presentation much more complex to include the estimation of $h$ and $b$ in the paper?

---

### Official Review · AnonReviewer4 · 2020-10-27
**Solid contribution on point processes and variational inference**

**Rating:** 7
**Confidence:** 2

**Review:**

The paper provides a stochastic variational inference method for
approximating posterior path measures for doubly-stochastic Poisson processes
conditioned on realisations of paths of the Poisson process. The intensity
process is modelled as the solution to a diffusion stochastic differential
equation. The authors compare their method experimentally to the numerical
solution of the associated system of stochastic partial differential
equations using the finite element method.

The paper, although technically difficult, is well written and clearly
presented. While I cannot validate the mathematical content in detail, the
paper seems technically correct. It deals with a well-defined problem in a
difficult mathematical setting.

The idea of using variational inference for approximating the smoothing
posterior density seems well-founded. The method is experimentally validated
on simulated and real data.

In summary, though my knowledge of point process theory is not sufficient to
evaluate all aspects of the paper in detail, I find the paper to be a solid
contribution worthy of publication at ICLR.

---

> ### Author Response · Authors · 2020-11-13
> **Thank you for your positive comments about our work.**
>
> We have uploaded the revised paper incorporating all the comments and a copy (as supplementary) with changes highlighted in blue. Please let us know if you have any questions or comments.

---

### Official Review · AnonReviewer1 · 2020-10-28
**The idea of the SDE-modulated cox is interesting, but the writing needs to be improved.**

**Rating:** 6
**Confidence:** 3

**Review:**

This paper proposes an interesting point process named diffusion modulated cox processes, which generalizes the stochastic intensity to a stochastic differential equation. The variational inference method looks sound.

Pros:
+ The generalization of SDE-type intensity is novel. The proposed stochastic variational inference makes sense. Especially the neural network solution is meaningful and will have an impact on the learning of point processes.
+ Good empirical performance and analysis.

Cons:
- I strongly recommend the authors to further improve the presentation of the current draft. Sometimes the notations are not consistent, like $\mathbb{E}$ and $E$, $\mathbb{R}$ and $R$. The definition of some of the notations are not very clear, such as Eq(3), what is the definition of the conditional expectation? Is it the expectation of the intensity function given all the information about the process?  Besides, please give a brief introduction of the mathematical background, such as the smoothing posterior. It would also be better if the authors can give some examples to illustrate the advantages of the proposed diffusion modulated cox process.
- Please discuss the influence of the non-negativity of the intensity function. The paper claims that $h$ is non-negative and thus can be an identity function, which is false.

---

> ### Author Response · Authors · 2020-11-13
> **Thank you for your valuable comments.**
>
> Please find our response below:
> 1.  Sometimes...... $\mathbb{E}$ and E, $\mathbb{R}$ and R \
> R. Thank you for pointing this out. We have gone through all the notations and ensured that they all are consistent.
>
> 2. The definition of some of the notations ........ about the process? \
> R.  Yes your understanding is correct. Precisely, by $\(\mathbb{E}[x_t|N_{0,T}]\)$, we mean $\(\mathbb{E}[x_t| \sigma(\mathbf{N}_u,0\leq u \leq T ) ]\)$, where  $\sigma(\mathbf{N}_u,0\leq u \leq T) $ is the smallest sigma algebra generated by count observations from time 0 to $T$. We have added a sentence after eq. (3) to explain this fact.
> 3. Besides, please give a brief ....... Cox process.\
> R.  We have added a paragraph on page 3 to briefly introduce smoothing posterior with additional references. Moreover on page 3, we have also added examples with references to illustrate the importance of diffusion modulated Cox processes in modeling various service and biological systems.
> 4. Please discuss the ...... is false.\
> R. Thank you for pointing this typo out. We meant that ‘h’ cannot be an identity function. We have fixed that statement. Moreover, the intensity function has to be non-negative as per the definition of the Cox process, therefore mapping $h$ has to be non-negative. We have added a sentence after eq. (1) for clarity.
>
> Note: We have uploaded the revised paper incorporating all your comments and a copy (as supplementary) with changes highlighted in blue.

---

### Decision · Program_Chairs · 2021-01-07
**Final Decision**

**Decision:**

Reject

**Comment:**

The paper presents a stochastic variational inference method for posterior estimation in a Cox process with intensity given by the solution to a diffusion stochastic differential equation. The reviewers highlight the novelty of the approach. Some of the concerns with regards to clarity have been addressed by the authors satisfactorily.

However, an important issue of the approach is that of estimating model parameters, which the authors do not address explicitly by simply referring to that as the task of the modeller. I believe this is an important issue and, although some of the parameters can be estimated along with the neural network parameters, this has not been shown empirically. Along a similar vein, the paper only presents results on a single real dataset (the bike-sharing dataset), which questions the applicability of the approach and no other baseline method is presented. At the very least, the authors should have provided an objective evaluation to other doubly stochastic point process models, e.g. based on Gaussian processes, where modern stochastic variational inference algorithms have been presented.